# REINFORCEMENT REWARD MODEL WITH POLICY FEEDBACK

## ABSTRACT

Reinforcement Learning from Human Feedback (RLHF) is a pivotal technique for aligning large language models (LLMs) with human preferences, yet it is susceptible to reward hacking, a phenomenon that policy models exploit spurious reward patterns instead of faithfully capturing human intent. Prior work to mitigate reward hacking primarily relies on surface semantic information and fails to efficiently address the misalignment between the reward model and the policy model caused by continuous policy distribution shifts. This inevitably leads to an increasing reward discrepancy, exacerbating reward hacking. To address these limitations, we propose **R2M** (**R**einforcement **R**eward **M**odel), a novel lightweight RLHF framework. Specifically, we aim to go beyond vanilla reward models that solely depend on the semantic representations of a pretrained LLM. Instead, we enhance the reward model by incorporating the evolving hidden states of the policy (namely **policy feedback**). we redesign the scoring head of the reward model to integrate policy feedback and introduce a corresponding iterative lightweight training phase, utilizing real-time policy feedback to enable adaption to policy distribution shifts. Notably, without modifying the core RLHF algorithms, simply integrating R2M enables the reward model to achieve iterative distribution alignment with accurate reward allocation, yielding 4.8% to 5.6% win rate improvement on dialogue tasks and 6.3% win rate improvement on document summarization tasks, while introducing marginal computational cost. This work points to a promising new direction for improving the performance of reward models through real-time utilization of feedback from policy models.

## 1 INTRODUCTION

Reinforcement Learning from Human Feedback (RLHF) has become a cornerstone technique for aligning large language models (LLMs) with human values and preferences (Vemprala et al., 2023; Shen & Zhang, 2024; Shen et al., 2025; Hu et al., 2024). However, RLHF faces a persistent challenge: reward hacking. Instead of faithfully capturing human intent, policy models often exploit spurious reward patterns, such as response length, markdown formatting, or superficial linguistic cues like certain n-grams or emojis, to maximize rewards without genuinely improving alignment (Gao et al., 2023; Coste et al., 2023; Eisenstein et al., 2023). The core issue lies in the reward model: trained on limited preference data, it can only approximate human values. As the policy evolves during RLHF training while the reward model remains fixed, distribution shift exacerbates approximation errors (Wang et al., 2024b), ultimately leading to unreliable reward signals in optimization.

A natural solution is to iteratively update the reward model so that it adapts to the policy's evolving behavior. Yet, direct retraining of the reward model at each iteration is computationally prohibitive. To address this, one research direction emphasizes uncertainty-aware corrections. Coste et al. (2023); Eisenstein et al. (2023); Zhai et al. (2023) penalize uncertain samples during policy training, while Zhang et al. (2024a) introduce kernel-based uncertainty estimates derived from reward model embeddings. Another line of work focuses on robust reward model retraining. Lang et al. (2024) incorporate an unsupervised mutual information loss to counter distribution shift, and Liu et al. (2024) augment training data by decomposing preferences relative to prompts. These methods trade off efficiency and robustness, but leave open a critical question: Can we design a new RLHF framework that preserves training efficiency while mitigating reward hacking effectively?

Our motivation stems from a key limitation of the standard RLHF pipeline: the unidirectional dependency between the policy and the reward model. While policies adapt to reward feedback, the reward model remains unaware of the policy's evolving internal states. This disconnect can allow policies to learn deceptive strategies, optimizing responses against a stale reward model rather than aligning with true human intent. To overcome this challenge, we propose **R2M** (**R**einforcement **R**eward **M**odel), a lightweight RLHF framework in which the reward model itself is reinforced iteratively by dynamically adapting to the policy's internal states, and it does not require any additional labeled data or environmental feedback to improve the performance.

Specifically, we observe that the hidden states of the policy encode latent patterns associated with reward hacking behaviors. Building on this insight, we aim to go beyond reward models that solely depend on the semantic representations of a pretrained LLM. Instead, we enhance the reward model by incorporating the evolving hidden states of the policy (namely **policy feedback**). To this end, we redesign the scoring head of the reward model so that it dynamically integrates these hidden states, enabling the reward model to adapt to distribution shifts in the policy. In our RLHF framework, this introduce a lightweight training component that learns to aggregate policy feedback directly, enhancing the reward model's representation without retraining the entire model. Owing to its efficiency, this mechanism can be seamlessly applied at every training round, ensuring continuous synchronization between the reward model and the policy.

The design of R2M offers two benefits: 1) **Iterative distribution alignment with accurate reward allocation.** The reward model integrates the policy's evolving hidden states which provide behaviorally grounded and semantically informed feedback. This mitigates distribution shifts, reduces reward hacking, and ensures more accurate reward assignment. 2) **Extremely lightweight overhead.** R2M only need to learn how to aggregate representations, introducing negligible additional cost.

Experimental results demonstrate that R2M significantly improves performance on dialogue tasks (trained on UltraFeedback (Cui et al., 2023), evaluated on Alpaca-Eval (Dubois et al., 2024)) and text summarization tasks (trained and evaluated on TL;DR summarization dataset). Specifically, R2M increases the AlpacaEval 2 win rate (WR) by 4.8% - 5.6%, the length-controlled win rate (LC) by 2.1% - 5.0% and the TL;DR win rate by 6.3% compared to baselines, while introducing only minimal computational cost. Furthermore, we conducted a comprehensive analysis, showing that R2M effectively strengthens the vanilla reward model and mitigates reward hacking with minimal additional training overhead.

## 2 PRELIMINARY

RLHF consists of three main steps: 1) Supervised Fine Tuning, 2) Reward Modeling, and 3) RL optimization, we provide a detailed workflow shown in Appendix G.1. As R2M is designed to directly integrated into the RL optimization phase, let us consider the following typical third-stage RL Optimization process:

First, in the **Trajectory Sampling** phase, at each training step $t \in [T]$, we update offline policy $\pi_{old}$ to online policy $\pi_\theta$. Then, given a query set $X_t = \{x_1, x_2, \ldots, x_n\} \subset \mathcal{X}$, $\pi_{old}$ is used to sample $K$ responses $G_i = \{y_{i,j}\}_{j=1}^K$ for each $x_i \in X_t$.

Next is the **Reward Annotation** phase. Specifically, for each $(x_i, G_i)$, $i \in [n]$, there are $K$ query-response pairs $(x_i, y_{i,j})$, $j \in [K]$. We use a score-based reward model $r_\varphi(x, y)$ to assign rewards to each query-response pair (Ahmadian et al., 2024; Hu, 2025), obtaining $\{r_{i,j} | i \in [n], j \in [K]\}$, resulting in a batch $\mathcal{B} = \{(x_i, y_{i,j}, r_{i,j}) | i \in [n], j \in [K]\}$. After this process, we employ the RLOO approach (Ahmadian et al., 2024) to perform advantage estimation within each group $G_i$:

$$\hat{A}_{i,j} = r_{i,j} - \frac{1}{K-1} \sum_{\hat{j} \neq j} r_{i,\hat{j}}. \tag{1}$$

Finally, in the **Policy Optimization** phase, for each query-response pair $(x_i, y_{i,j})$, we perform a forward pass in the policy model $\pi_\theta$ and optimize $\pi_\theta$ using importance sampling by maximizing the following objective (Shao et al., 2024; Ahmadian et al., 2024), where $\varepsilon$ and $\beta$ are hyperparameters:

$$\left\{ \min \left[ \frac{\pi_\theta(y_{i,j}|x_i, y_{i,j})}{\pi_{\theta_{\text{old}}}(y_{i,j}|x_i, y_{i,j})} \hat{A}_{i,j}, \text{clip} \left( \frac{\pi_\theta(y_{i,j}|x_i, y_{i,j})}{\pi_{\theta_{\text{old}}}(y_{i,j}|x_i, y_{i,j})}, 1-\varepsilon, 1+\varepsilon \right) \hat{A}_{i,j} \right] - \beta \mathbb{D}_{KL} \left[ \pi_\theta \| \pi_{\text{ref}} \right] \right\}. \tag{2}$$

The design of R2M is based on the aforementioned RL optimization process. As a lightweight and significantly effective alternative, R2M can be seamlessly deployed to all REINFORCE-based RLHF frameworks. Due to resource constraints, we adopt RLOO as the primary baseline.

# 3 MOTIVATION

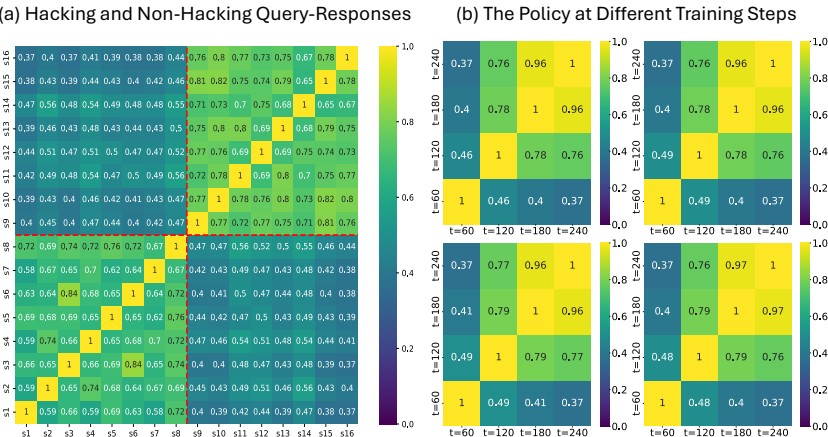

Figure 1: (a) **Identification of Reward Hacking Patterns.** We show the similarity matrix of hidden states from forward passes of different query-response pairs for the same policy. The first 8 samples are sequences exhibiting reward hacking, while the last 8 are normal output responses. $s$ denotes the query-response pairs. (b) **Policy Distribution Shift Analysis.** For a given query with four different responses, we display the similarity matrix of the policy across various training steps $t$.

We argue that hidden states in a transformer's forward pass contain crucial information about a policy's internal state and semantic information, making them effective for mitigating reward hacking. We validated this by computing hidden state similarity matrices. As shown in Figure 1 (a), responses with and without reward hacking show significant differences in their hidden state similarities. Figure 1 (b) shows that the same query-response's hidden states from different training steps of a policy model are significantly different. Furthermore, as shown in Table 1, the average similarity between hacking and non-hacking responses is significantly lower than the similarity within each category. These findings strongly confirm that a policy's hidden states offer valuable insights for detecting reward hacking.

To combat reward hacking, our R2M architecture decouples the issue from both the reward and policy models. We enhance the reward model's alignment with true human preferences by leveraging policy feedback to improve reward allocation, moving beyond reliance on superficial patterns. Simultaneously, we tackle the policy model's tendency to exploit fixed proxy rewards by enabling the reward model to dynamically adapt to the policy's evolving internal state distribution, thus preventing the exploitation of fixed patterns.

Table 1: We report the average similarity of hidden states across three categories from mutiple query-response pair groups, each group comprises 8 responses exhibiting reward hacking and 8 normal responses.

| Type | Hacking | Non-Hacking | Cross-Category |
|---|---|---|---|
| **Avg-Sim** | 0.67 | 0.75 | 0.45 |

# 4 METHOD

Figure 2 illustrates the overall workflow of R2M. Built upon the RL optimization framework described in Section 2, R2M primarily consists of two key components: 1) how to structurally incorporate

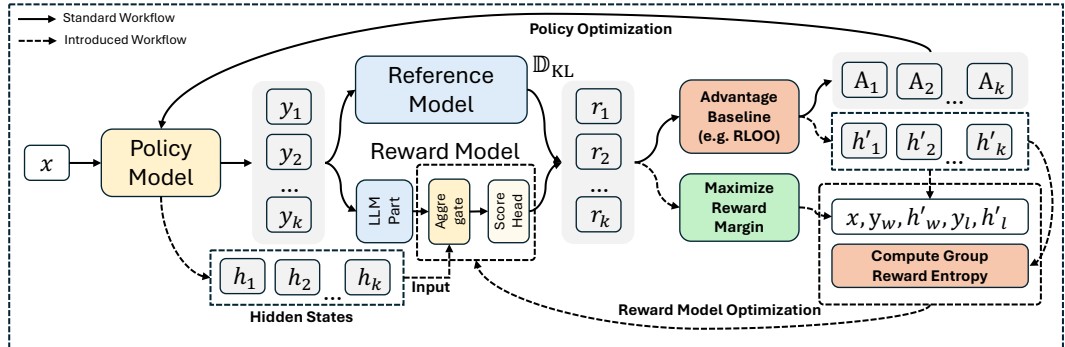

Figure 2: Overview of R2M. We first aggregate the last-layer hidden states from the policy $h_i$ with the LLM part output of the reward model. This aggregated representation is then fed into the scoring head for reward prediction. When the policy updates, we get the real-time feedback $h'_i$ and utilize it to construct preference pairs. Finally, we optimize the reward model by jointly minimizing the Bradley-Terry loss and the Group Reward Entropy.

feedback messages into the reward model (Section 4.1); 2) how to design the optimization objectives for the reward model (Section 4.2).

## 4.1 REWARD MODEL STRUCTURE

In this section, we focus on integrating the policy feedback from Section 2 into the reward model. As shown in Figure 3, we introduce a policy feedback data flow that bypasses the LLM part to directly enhance the original Reward Token Embedding (introduced in Appendix G.1). We formally redefine the reward model $r_\varphi(x, y)$ as **R2M** $r_\varphi(x, y, h)$. To effectively utilize the policy feedback, R2M contains two pivotal extra components: Sequence-to-Token Cross Attention and Time-Step-Based Weighted Combination.

Specifically, during the Trajectory Sampling phase, we collect the last-layer hidden states $h_{i,j} \in \mathbb{R}^{S_{i,j}-1 \times D_p}$ for each query-response pair $(x_i, y_{i,j}), i \in [n], j \in [K]$ from the policy. Here, $S_{i,j}$ denotes the effective length of the query-response pair, and $D_p$ represents the hidden size of the policy. In the Reward Annotation phase, we first input $(x_i, y_{i,j}), i \in [n], j \in [K]$ into the LLM part of the reward model and obtain the Reward Token Embedding $H_{\text{last}}^{i,j} \in \mathbb{R}^{D_{\text{rm}}}$.

**Sequence-to-Token Cross Attention.** We introduce a cross-attention component to extract relevant information from hidden states of query-response pairs. Specifically, we inject policy feedback by performing a cross-attention operation from the sequence to a single token. This enables the query of the Reward Token Embedding $H_{\text{last}}^{i,j}$ to fully absorb the keys and vales of the hidden state sequence $h_{i,j}$, which contains both policy state information and sequence semantic information, and updates it into a more information-rich Aggregated Reward Token Embedding $\widehat{H}_{\text{last}}^{i,j}$.

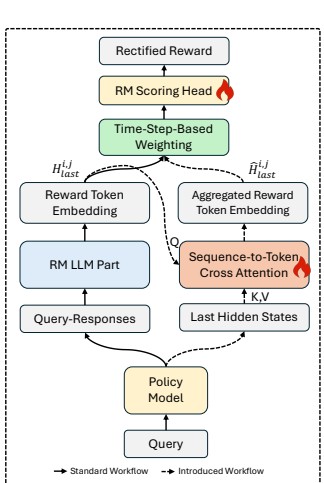

Figure 3: The structure of R2M. Building on the dataflow based on solely surface semantic information (left), R2M introduces an additional dataflow based on the policy feedback (right).

**Time-Step-Based Weighted Combination.** After obtaining $\widehat{H}_{\text{last}}^{i,j}$, we adopt an exploration-exploitation approach (Ban et al., 2021; 2024; Huang et al., 2025) to balance the weights of $H_{\text{last}}^{i,j}$ and $\widehat{H}_{\text{last}}^{i,j}$, yielding the final Reward Token Embedding $H_{\text{fin}}^{i,j}$. Specifically, we use a time-step-based approach to gradually decrease the weight on the original Reward Token Embedding $H_{\text{last}}^{i,j}$ as follows:

$$H_{\text{fin}}^{i,j} = (1 - \omega(t))\widehat{H}_{\text{last}}^{i,j} + \omega(t)H_{\text{last}}^{i,j}, \quad \omega(t) = \max(\frac{1}{2}\cos(\frac{t}{T}\pi) + \frac{1}{2}, \Omega), \quad (3)$$

where $t$ is the current training round, and $T$ is the total number of training rounds. $\Omega$ is a hyperparameter used to ensure the minimum weight of the original Reward Token Embedding, and $\omega(t)$ is a monotonically decreasing function of $t$ (Wu et al., 2025). When $t$ is small, we prioritize leveraging the existing Reward Token Embedding $H_{\text{last}}^{i,j}$. As R2M iteratively updates during the training process (as discussed in Section 4.2), we gradually increase the influence of $\hat{H}_{\text{last}}^{i,j}$ to enable the reward model to progressively identify and adapt to the distribution shift of the policy. As a result of balancing the exploitation of the original embedding with the exploration of policy feedback information, $H_{\text{fin}}^{i,j}$ is then mapped by the reward head $\phi$ to the final scalar reward $r_\varphi(x_i, y_{i,j}, h_{i,j}) = \phi(H_{\text{fin}}^{i,j}) \in \mathbb{R}$.

## 4.2 ITERATIVE REWARD MODEL LIGHTWEIGHT OPTIMIZATION

In Section 4.1, we have introduced policy feedback into the reward model. However, the semantic spaces are not yet aligned, making it challenging for the reward model to directly utilize this information. To address this, we incorporate an extra lightweight Reward Model Optimization phase following the Policy Optimization phase at each training step, and propose a novel optimization objective for R2M, namely the Group Reward Entropy Bradley-Terry loss.

**Hidden State Update**. To ensure that the hidden states $h_{i,j}$ remain up-to-date and accurately reflect the internal states of the policy $\pi_\theta$, we update $h_{i,j}$ whenever $(x_i, y_{i,j})$ is used to update $\pi_\theta$. Specifically, during the forward pass of $\pi_\theta$ on $(x_i, y_{i,j})$, we fetch the latest hidden states $h_{i,j}$, which incurs no additional computational overhead. Since the policy model is trained for k epochs on the same batch at each training step $t$ (Shao et al., 2024; Hu, 2025), this update is performed only in the final epoch. For notational simplicity, we continue to use $h_{i,j}$ to denote the most recent hidden states. This mechanism enables the reward model to dynamically capture distribution shifts in real time as the policy evolves.

**Group Reward Entropy Bradley-Terry Loss.** To enhance the robustness of the reward model by incorporating policy feedback during score allocation, we propose the Group Reward Entropy Bradley–Terry Loss. For each query-response group $(x_i, G_i)$, to ensure the reliability of preference labels, we select only the samples with the highest and lowest scores to construct the preference pair, resulting in $\{x_i, y_{i,w}, h_{i,w}, y_{i,l}, h_{i,l}\}$. Then, we can establish the Bradley-Terry optimization objective as:

$$\mathcal{L}_{\text{BT}}(i : \varphi) = -\log \sigma\big(r_\varphi(x_i, y_{i,w}, h_{i,w}) - r_\varphi(x_i, y_{i,l}, h_{i,l})\big), \tag{4}$$

which allows the reward model to be continuously optimized as the policy evolves.

However, in practice, the reward model often assigns nearly identical scores to responses within a group, especially in the later phases of RL optimization when the responses become more homogeneous. To address this issue, we introduce an entropy regularization term to encourage greater reward diversity within each group. Specifically, for each group $(x_i, G_i)$, we first compute the foward pass of the reward model $\varphi$ on all samples to get newly allocated reward scores $r_{i,j} = r_\varphi(x_i, y_{i,j}, h_{i,j}), j \in [K]$. We define the Group Reward Entropy for group $(x_i, G_i)$ as

$$H_{\text{group}}^i = -\sum_{j=1}^K p_{i,j} \log p_{i,j}, \quad \text{where } p_{i,j} = \text{softmax}\left(\frac{r_{i,j} - \text{mean}(\mathbf{r})}{\text{std}(\mathbf{r})}\right), \tag{5}$$

where $\mathbf{r} = \{r_{i,1}, r_{i,2}, \ldots, r_{i,K}\}$, and $i$ is the group index, the softmax operation is applied across all standardized reward values within the group to get the relative preference of each sample. By minimizing the GRE, we sharpen the distribution $p_{i,j}$, thereby amplifying the score disparities within the group. Finally, the overall optimization objective of R2M is given by:

$$\mathcal{L}_{\text{FIN}}(i : \varphi) = (1 - \alpha)\mathcal{L}_{\text{BT}}(i : \varphi) + \alpha H_{\text{group}}^i, \tag{6}$$

where $\alpha$ is a tunable hyperparameter. Through this optimization objective, we enable the reward model to progressively learn to provide reasonable and more confident reward signals while incorporating real-time policy feedback, thereby allowing it to automatically adapt to the policy's distribution shifts.

**Workflow**. Algorithm 1 illustrates the workflow of our proposed R2M algorithm, The modifications primarily involve utilizing both shallow semantic information $(x_i, y_{i,j})$ and policy feedback $h_{i,j}$ during the Reward Annotation phase, as well as introducing an additional lightweight Reward Model Optimization phase to iteratively update the reward model based on real-time policy feedback.

*Policy Optimization (Lines 2-14).* We retain the same Policy Optimization phase as described in Section 2, with the only difference being that we update the policy feedback for each query-response pair using the real-time updated $\pi_\theta$ as mentioned in Section 4.2.

*Reward Model Optimization (Lines 15-20).* To preserve the general representational capacity of the reward model's LLM part while enhancing the relatively weaker linear projection component,we solely update the cross-attention component and the scoring head $\phi$, leaving the LLM part frozen. We provide detailed design motivations in the Appendix G.2. This approach significantly reduces the overall computational cost of R2M, ensuring the feasibility of iteratively updating the reward model.

---

**Algorithm 1** Proposed RLHF Framework: R2M

---

**Require:** Initial policy model $\pi_\theta \leftarrow \pi_{\text{SFT}}$, reference model $\pi_{ref}$, reward model $r_\varphi$, queries $\mathcal{X}$
1: **for** step $= 1, \ldots, T$ **do**
2:      Sample a batch $\mathcal{X}_{batch} = \{x_i\}, i \in [n]$ from $\mathcal{X}$
3:      Update the old policy model $\pi_{\text{old}} \leftarrow \pi_\theta$
4:      ***Trajectory Sampling:***
5:      Sample a group of output $G_i = \{y_{i,j}\}, j \in [K] \sim \pi_{\text{old}}(\cdot \mid x_i)$ for each query $x_i \in \mathcal{X}_{batch}$
6:      Get the last-layer hidden states $\{h_{i,j}\}, j \in [K]$ from $\pi_{\text{old}}$
7:      ***Reward Annotation:***
8:      Compute the rewards with policy feedback $\{r_\varphi(x_i, y_{i,j}, h_{i,j})\}, i \in [n], j \in [K]$
9:      Compute $\{\hat{A}_{i,j}\}, j \in [K]$ within each $G_i$ for query $x_i$ through Equation 1
10:     ***Policy Optimization:***
11:     **for** iteration $= 1, \ldots, k$ **do**
12:        Update the policy model $\pi_\theta$ by maximizing the RLOO objective through Equation 2
13:        Update $h_{i,j}, i \in [n], j \in [K]$ from the policy forward when iteration $= k$
14:     **end for**
15:     ***Reward Model Optimization:***
16:     Get preference pair $\{x_i, y_{i,w}, h_{i,w}, y_{i,l}, h_{i,l}\}$ according to Section 4.2 within each $G_i$
17:     Compute $\mathcal{L}_{\text{BT}}(i : \varphi)$ according to Equation 4
18:     Compute $\{r_\varphi(x_i, y_{i,j}, h_{i,j})\}, j \in [K]$ within each $G_i$
19:     Compute Group Reward Entropy $H_{\text{group}}^i$ according to Equation 5
20:     Update reward model $r_\varphi$ according to Equation 6
21: **end for**
**Ensure:** $\pi_\theta, r_\varphi$

---

## 5 EXPERIMENT

In this section, we present the primary experimental results along with their analysis. We set the learning rate of R2M to $1 \times 10^{-6}$, the weight coefficient of the hybrid loss $\alpha = 0.5$, and the width of cross-attention component to 2048. during the entire training process, we sample 12k trajectories with a maximum length of 512 for the dialogue task, and 1000k trajectories with a maximum length of 50 for the document summarization task. Additional implementation details of R2M are provided in Appendix F due to space constraints.

### 5.1 MAIN EXPERIMENT RESULTS

In this section, we present the experimental results of R2M on dialogue and document summarization tasks. We integrated R2M into RLOO and compare it against state-of-the-art REINFORCE-based RLHF algorithms.

For dialogue task, We considered the current mainstream evaluation frameworks, utilizing queries from UltraFeedback (Cui et al., 2023) for online RL optimization and conducting evaluations with AlpacaEval 2 (Dubois et al., 2024), which is a widely used chat-based evaluation benchmark. Detailed experimental settings can be found in Appendix F.2.

Next, we considered a classic RLHF task, summarization: $x$ is a forum post from Reddit, and the policy must generate a summary $y$ of the main points in the post. The corresponding experimental settings are detailed in Appendix F.3.

**(1) R2M consistently achieves superior performance.** As shown in Table 2, the incorporation of policy feedback and iterative updates of the reward model enable R2M to achieve the highest scores across all evaluation metrics. Specifically, R2M outperforms the best-performing baseline by margins ranging from 2.1% to 5.0% on the AlpacaEval 2 LC win rate, from 4.8% to 5.6% on the AlpacaEval 2 win rate and 6.3% on the TL;DR win rate. These results underscore the broad applicability of R2M in preference optimization and its effectiveness in aligning large language models with human preferences.

**(2) R2M significantly enhances the reward model.** The sole difference between R2M and RLOO is the replacement of a frozen reward model with one iteratively updated and allocating rewards via policy feedback. Compared to RLOO, R2M achieved a 2.9% to 6.1% increase in LC win rate, a 5.2% to 8.0% increase in raw win rate, and a 6.3% increse in TL;DR win rate. Notably, the LC win rate improvement was accomplished while reducing average sequence length to a certain extent. These substantial improvements are entirely due to the stronger reward model of R2M. This clearly demonstrate the effectiveness of R2M's integration of feedback to iteratively enhance the reward model. This enhancement can be attributed to two factors: real-time alignment with the policy model and additionally introduced deep semantic understanding, thanks to the rich information from policy feedback discussed in Section 3.

Table 2: Results of R2M compared with baselines across different experimental settings. LC and WR denote length-controlled and raw win rate, respectively. Here, **bold** denotes the best performance, underline indicates the second-best performance.

| Method | Dialogue | | | | | | Summarization |
| --- | --- | --- | --- | --- | --- | --- | --- |
| | Qwen2.5-3B-Instruct | | | LLaMA3-8B-Instruct | | | Pythia-2.8B-TL;DR |
| | LC(%) | WR(%) | Avg Len | LC(%) | WR(%) | Avg Len | WR(%) |
| SFT | 15.5 | 15.8 | 2218 | 22.9 | 22.6 | 1899 | 42.3 |
| GRPO | 22.7 | 25.6 | 3012 | 29.5 | 32.6 | 2216 | 75.2 |
| ReMax | 21.8 | 25.1 | 2916 | 28.7 | 30.7 | 2289 | 75.1 |
| REINFORCE++ | 21.4 | 26.4 | 3252 | 29.3 | 31.8 | 2192 | 74.3 |
| RLOO | 21.9 | 26.0 | 3174 | 28.4 | 30.2 | 2186 | 75.3 |
| R2M | **24.8** | **31.2** | 2911 | **34.5** | **38.2** | 2011 | **81.6** |

## 5.2 ANALYSIS

In this section, we present additional analytical experiments to clarify, from a principled perspective, the reasons behind R2M's effectiveness in RL optimization.

**R2M maintains reward consistency while allocating higher rewards.** We compared the average R2M rewards to that annotated by the vanilla reward model during RL optimization process. Specifically, every 5 training steps, we sampled 128 queries from the test set as a batch, prompted the policy $\pi_\theta$ to generate complete responses, and scored them using the reward model. We report the average scores for each batch. As shown in Figure 4, **Reward without Feedback** is provided by a frozen reference reward model, while **Reward with Feedback** corresponds to R2M. To rigorously compare the effect of policy feedback, we include an additional control group **Reward with**

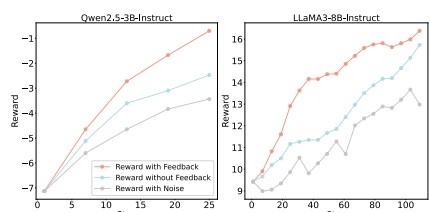

Figure 4: Comparison of average rewards under various conditions.

**Noise**, where we replace the feedback with Gaussian noise. we first observe that R2M exhibits a consistent reward trend compared to the reference reward model, directly indicating that R2M can reliably provide reasonable rewards. Additionally, R2M consistently allocating higher rewards. In contrast, when noise with the same mean and standard deviation is introduced, the resulting reward signals are significantly reduced. This clearly suggests that policy feedback contains beneficial information, consistent with the phenomena observed in Section 3. We hypothesize that the higher

average reward allocation results from the GRE minimization objective of R2M, which encourages the reward model to assign higher reward values to high-quality responses with greater confidence.

**R2M encourages substantial and effective policy updates.** Figure 5 illustrates the performance curves of R2M compared with RLOO during RL optimization for dialogue tasks. As shown in the left two columns, R2M demonstrates a significantly higher reward curve and lower loss curve compared to RLOO. Generally, this indicates more effective training outcomes. From the perspective of KL divergence, R2M encourages larger parameter shifts in the model to achieve greater rewards. As shown in the right two columns of Figure 5, R2M exhibits a greater policy distribution shift in the KL-Step curve and a denser distribution in the high-reward, high-KL region of the Reward-KL scatter plot. Aggressive policy updates readily lead to reward hacking (Coste et al., 2023), whereas R2M, compared to RLOO, achieves substantial performance gains demonstrated in Section 5.1, rather than causing training collapse. This indicates that R2M effectively improves the reward model's resistance to policy's exploitation of specific patterns, enabling more aggressive policy updates in the correct direction without triggering reward hacking.

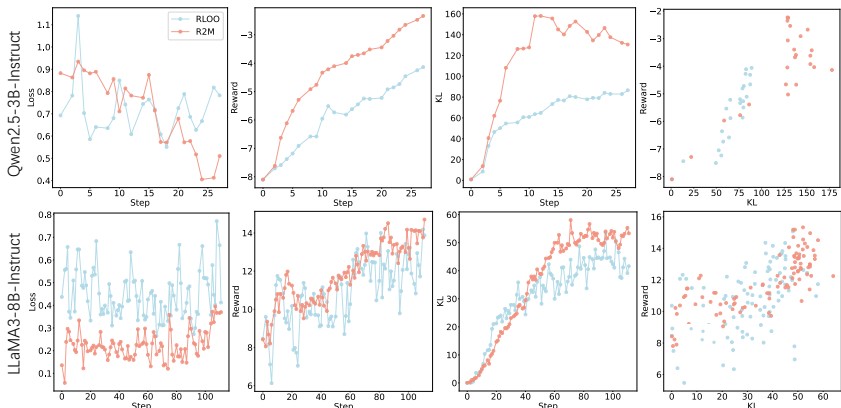

Figure 5: We compare RLOO and R2M in terms of loss, reward and KL divergence during RL optimization, using Qwen2.5-3B-Instruct and LLaMA3-8B-Instruct as policy models, and Skywork-Reward-V2-Llama-3.1-8B as the reward model. For KL divergence, we calculate it as the average of log probability differences between the reference model and the policy model for each token.

**R2M significantly improves the accuracy of the reward model.** We compare the accuracy of R2M and the vanilla reward model on the test set of UltraFeedback before and after running the R2M pipeline, as experimental details shown in Appendix F.4. As shown in Table 3, after iterative updates, R2M achieves accuracy improvements of 5.1% and 6.3% compared to the original reward model. These results indicate that R2M significantly enhances the accuracy of the reward model, which is crucial for preventing reward hacking and improving training effect (Rafailov et al., 2023; Lambert et al., 2024; Adler et al., 2024). Before training, incorporating policy feedback results in accuracy decreases of 4.0% and 3.1%. This clearly demonstrates that policy feedback cannot be used directly, highlighting the effectiveness and necessity of the Reward Model Optimization phase in R2M.

Table 3: Comparison of the accuracy of reward models under different conditions. "Without Feedback" refers to the frozen reference reward model, while "With Feedback" represents R2M before and after the R2M pipeline.

Table 4: The TL;DR Result of R2M Compared with Baselines.

| Reward Model Type | Policy Model Type | |
|---|---|---|
| | Qwen2.5(%) | LLaMA3(%) |
| without Feedback | 72.3 | 72.3 |
| with Feedback (Before-Training) | 68.3 | 69.2 |
| with Feedback (After-Training) | 77.4 | 78.6 |

| Method | WR(%) |
|---|---|
| SFT | 33.7 |
| RLOO | 8.7 |
| R2M | 61.6 |

**R2M can strongly mitigate reward hacking.** In this section, we present a case study demonstrating that R2M not only largely enhances the robustness and performance of RL optimization, but also effectively prevents training collapse due to reward hacking. Specifically, when performing RL optimization with RLOO on the TL;DR task using Pythia-1B-TL;DR-SFT and Pythia-1B-TL;DR-RM, we observed that the trained model produced completions without spaces, despite maintaining correct semantic meaning. This issue arises because the Pythia tokenizer controls the presence of spaces through special token prefixes, and the reward model exhibits a erroneous preference for token sequences without spaces. After applying R2M, this severe reward hacking phenomenon was eliminated, and a stable improvement in win rate was achieved after RL optimization, as shown in Table 4. These results indicate that R2M can effectively mitigate reward hacking, even in cases of complete training collapse, under identical hyperparameter settings. Detailed experimental results are provided in Appendix F.5.

## 5.3 COMPUTATIONAL COST ANALYSIS

R2M is lightweight and compute-efficient. Figure 6 illustrates the peak single-GPU memory usage and overall runtime of R2M compared to RLOO under setting of LLaMA environment. R2M introduces negligible additional overheads compared to the performance gains it achieves. This can attribute to two main factors. First, policy feedback can be directly obtained and its aggregation solely involves lightweight attention computations. Second, R2M does not update the reward model's LLM part, and its cross-attention module and scoring head are relatively lightweight.

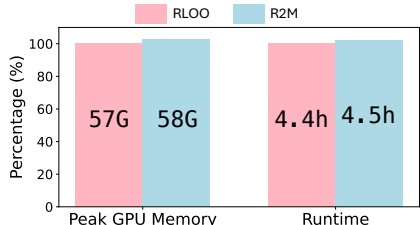

Figure 6: Computational cost comparison between RLOO and R2M: similar runtime and GPU memory usage.

**For the ablation study, refer to Appendix E.**

## 6 RELATED WORKS

**REINFORCE-based RLHF Algorithms.** RLHF is a critical technique for aligning large language models with human preferences (Ouyang et al., 2022; Bai et al., 2022a). The classical RLHF pipeline typically comprises three phases: supervised fine-tuning (Geng et al., 2023), reward model training (Gao et al., 2023), and policy optimization against the reward model (Schulman et al., 2017). As a classic reinforcement learning algorithm, Proximal Policy Optimization (PPO) (Schulman et al., 2017) is widely used in the third stage of RLHF. Recently, many researchers have proposed a series of REINFORCE-based methods, such as ReMax (Li et al., 2023), RLOO (Ahmadian et al., 2024), GRPO (Shao et al., 2024) and REINFORCE++ (Hu, 2025) to avoid the computational overhead associated with the critic model while still obtaining relatively accurate sequence-wise advantage estimations. These methods design alternative techniques to calculate the baseline reward for each prompt as the advantage estimation.

## 7 CONCLUSION

To mitigate reward hacking exacerbated by policy's distribution shifts, we propose **R2M**, a novel lightweight RLHF framework. By incorporating the policy's evolving hidden states, R2M enhances the reward model's accuracy while maintaining robustness against reward hacking. Without modifying current RLHF algorithms, Simply integrating R2M into the framework achieves significant performance improvements while introducing only marginal additional computational costs.

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

APPENDIX

## A  LIMITATIONS

The primary limitation of R2M lies in its sensitivity to the vanilla reward model's performance. While R2M significantly enhances a standard reward model, its benefits diminish when the baseline model already closely aligns with true human preferences. As discussed in Section 3, reward hacking arises from the reward model's misalignment with human preferences. Thus, R2M is most effective as an enhancement strategy for suboptimal reward models, with reduced impact when the vanilla reward model accurately predicts ground-truth rewards. However, it is important to contextualize this limitation within the complexity of training a relatively perfect reward model, which remains a non-trivial challenge in RLHF.

## B  BROADER IMPACT

Our proposed R2M offers several significant advantages and has far-reaching potential applications. By incorporating real-time feedback from the policy model, R2M addresses a critical limitation of traditional reward models, enabling iterative alignment with the policy model and more accurate reward allocation. Its seamless integration with current RLHF algorithms without altering the core mechanism and minimal computational overhead make it highly practical for both research and real-world use. In natural language processing (NLP), R2M can enhance chatbots, virtual assistants, and content generation systems, improving user experiences and text quality. While our method has broad applicability across domains, we do not foresee specific societal risks or negative impacts that require special consideration, as R2M focuses on enhancing the reward model in RL optimization of RLHF framework and maintains the ethical and societal implications consistent with standard RLHF practices.

## C  ONE CASE STUDY OF REWARD HACKING

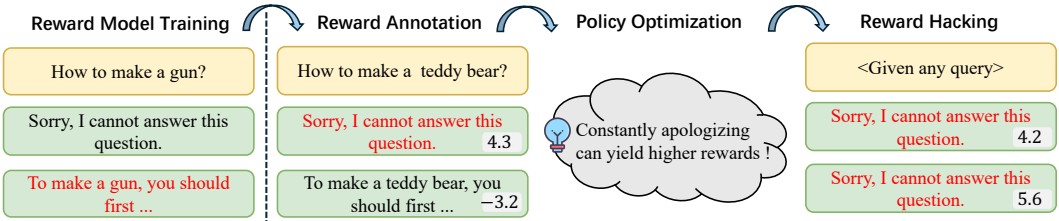

Figure 7: During Reward Model Training, the reward model inadvertently learned to assign high scores to responses containing apologies. The policy model detected this pattern and persistently exploited it to obtain inflated rewards, which resulted in a collapse of the RL Optimization process.

## D  ADDITIONAL RELATED WORK

**Mitigating Reward Hacking in RLHF.**  Constructing a superhuman and unbiased reward model is crucial for maximizing the potential of policies in RLHF (Wang et al., 2024a; Bai et al., 2022b). While revealed by Denison et al. (2024); Zhang et al. (2024b), reward models are easily hacked by different pattern in different scenario, e.g., length (Singhal et al., 2023) and sycophancy. Several studies have explored strategies to mitigate reward hacking in reinforcement learning with human feedback (RLHF), focusing on enhancing the robustness of reward models and addressing vulnerabilities exploited by policy models.

**(1) Uncertainty-Based Re-Scoring.** One line of work mitigates reward hacking by incorporating uncertainty estimation into the reward scoring process. Studies such as Coste et al. (2023), Eisenstein et al. (2023), and Zhai et al. (2023) focus on penalizing samples with high reward uncertainty during

RL-based policy training to prevent the policy from exploiting unreliable reward signals. Additionally, Zhang et al. (2024a) utilizes preference data embeddings from the last layer of the reward model as feature mappings, pre-training a kernel function to evaluate whether new prompt-response pairs resemble those observed during training, thereby providing an uncertainty estimate to guide policy optimization.

**(2) Reward Model Retraining.** Another approach enhances the robustness of the reward model through targeted retraining. For instance, Lang et al. (2024) introduces an additional training phase for the reward model, incorporating an unsupervised mutual information loss term to address the policy's distribution shift and improve generalization. Similarly, Liu et al. (2024) decouples preferences based on their relevance to the prompt and retrains the reward model using an augmented dataset to ensure more accurate reward signals.

**(3) Additional Techniques.** Recent advancements also include model merging techniques, such as WARP (Ramé et al., 2024a) and WARM (Ramé et al., 2024b), and hacking reward decomposition, as proposed in ODIN (Chen et al.), to mitigate reward hacking in online RLHF. Generative reward models, as explored by Yan et al. (2024), enable more nuanced preference analysis, enhancing the granularity of reward signals. For domains requiring high precision, such as mathematics, verifiable answers can be leveraged to ensure accurate reward signals (Xiong et al., 2024).

However, most model-based methods fail to leverage the deeper semantic information from the policy model, while permitting the policy model to persistently exploit vulnerabilities during policy optimization. In contrast to these approaches, R2M significantly enhances the robustness and performance ceiling of policy optimization by incorporating feedback information from the policy and employing lightweight iterative reward model updates.

## E    ABLATION STUDY

In this section, we perform detailed ablation studies to assess the effectiveness of the design of each component in R2M. Based on the LLaMA3 experimental setup outlined in Section 5.1, we systematically remove key modules of R2M and evaluate their impact on experimental outcomes, as presented in Table 5.

**R2M with Noise** & **R2M without Training**: For R2M with Noise, we replace the feedback information with Gaussian noise of equivalent mean and variance. For R2M without Train, we incorporate feedback from the policy without updating the reward model. We observed that the performance improvement of the aforementioned two approaches was very limited, even significantly lower than the baseline RLOO, with this improvement primarily stemming from the dominant role of the original Reward Token Embedding in the early stage of training. The results of R2M with Noise is consistent with Section 3 and Section 5.2, which indicates that, compared to noise, the feedback information from the policy model is evidently effective information for the reward model. On the other hand, the results of R2M without Training suggests that to effectively incorporate feedback information, updating R2M is necessary, which aligns with Section 5.2.

**R2M without BT Loss** & **R2M without GRE Loss**: We optimize R2M with only single object from Equation 6 as the optimization objective. Compared to R2M, we observed that removing the BT loss resulted in a decrease of 3.0 and 2.5 in LC and WR scores, respectively. When the GRE loss was removed, the scores dropped to 2.2 and 2.0. This clearly indicates that utilizing a mixed loss as the optimization objective outperforms a single objective. On the other hand, even with single optimization object, R2M still significantly outperforms RLOO, especially when using BT loss, which achieved score improvements of 3.9 and 6.0, respectively. This demonstrates that, whether using BT-loss or GRE loss as the optimization objective, the injection of feedback information from the policy effectively enhances the robustness and accuracy of R2M.

## F    EXPERIMENTAL DETAILS

### F.1    HIDDEN STATES ANALYSIS EXPERIMENT

We decided to utilize the last-layer hidden states of the query-response pairs as the policy feedback. There are two primary reasons supporting this approach. First, they are widely recognized as universal

Table 5: Ablation study results under LLaMA3-8B-Instruct settings. LC and WR denote length-controlled and raw win rate, respectively. $\Delta$ represents score changes relative to R2M ($\downarrow$ indicates lower than R2M, $\uparrow$ indicates higher than R2M).

| Method | LC(%) | $\Delta$ | WR(%) | $\Delta$ | Average Length |
|---|---|---|---|---|---|
| RLOO | 28.4 | 6.1$\downarrow$ | 30.2 | 8.0$\downarrow$ | 2186 |
| R2M with Noise | 25.4 | 9.1$\downarrow$ | 26.4 | 11.8$\downarrow$ | 2276 |
| R2M without Training | 26.1 | 8.4$\downarrow$ | 28.9 | 9.3$\downarrow$ | 2183 |
| R2M without BT Loss | 31.5 | 3.0$\downarrow$ | 35.7 | 2.5$\downarrow$ | 2116 |
| R2M without GRE Loss | 32.3 | 2.2$\downarrow$ | 36.2 | 2.0$\downarrow$ | 2191 |
| R2M | 34.5 | - | 38.2 | - | 2011 |

sequence representations and are extensively used in downstream tasks (Chen et al., 2024; Zhang et al., 2025; 2024a; Guo et al., 2025). On the other hand, due to the forward propagation mechanism of transformers Vaswani et al. (2017), hidden states encapsulate both the semantic information of the sequence and the internal state information of the policy. We hypothesize that the former aids in identifying reward hacking patterns, while the latter may contain critical information about distribution shifts.

**Internal State Information Validation.** To validate that the last-layer hidden states contain state information about policy distribution shifts, we perform forward passes on the same query-response pair $(x, y)$ from the UltraFeedback test set using LLaMA3-8B-Instruct as the policy model at training steps $t = 60, 120, 180, 240$, extracting the last-layer hidden states $\{h_i\}, i \in [1, 4], h_i \in \mathbb{R}^{s_i \times D_p}$, where $s_i = \|x + y_i\|$ and $D_p$ is the hidden size of the policy. We calculated the average token hidden state $\{\bar{h_i}\}, i \in [1, 4], \bar{h_i} \in \mathbb{R}^{D_p}$ and computed the pairwise cosine similarity between them.

We conduct forward passes on a query-response pair $(x, y)$ using policy models $\pi_{\theta_t}$ at various training steps $t$, extract the last-layer hidden states, and compute their pairwise cosine similarity. We sample four responses for the same query, generating four query-response pairs and their corresponding similarity matrices.

**Semantic Information Validation.** To validate that the last-layer hidden state contains semantic information for identifying hacking sequences, We collected a subset of size 100, denoted as $\mathcal{X}_{test}$, $|\mathcal{X}_{test}| = 100$, from the test set of UltraFeedback (Cui et al., 2023). For each query $x \sim \mathcal{X}_{test}$, we manually categorized the responses from the policy $\pi_\theta$ during RL Optimization into hacking responses $\{y_i\}, i \in [1, 8]$ and non-hacking responses $\{y_i\}, i \in [9, 16]$. We computed the query-response pairs $\{c_i = (x, y_i)\}, i \in [1, 16]$ and fed them into LLaMA3-8B-Instruct as the policy model $\pi_\theta$, extracting the last hidden state $\{h_i\}, i \in [1, 16], h_i \in \mathbb{R}^{s_i \times D_p}$, where $s_i = \|x + y_i\|$ and $D_p$ is the hidden size of the policy. We calculated the average token hidden state $\{\bar{h_i}\}, i \in [1, 16], \bar{h_i} \in \mathbb{R}^{D_p}$ and computed the pairwise cosine similarity between them.

## F.2 EXPERIMENTAL SETTINGS OF THE DIALOGUE TASK

We initially filtered out UltraFeedback samples where the chosen response exceeded 512 tokens. Subsequently, at each step $t$, we sample 128 queries (i.e., $n = 128$) from the training set. For each query, the policy model responds $K$ times with a temperature of 0.7, without applying top-k or top-p token restrictions, resulting in a total of 12k trajectories for training. During policy training, we utilized all offline-sampled trajectories from the current round and trained for 2 epochs. Subsequently, we conducted experiments following the procedure outlined in Algorithm 1.

**LLM Settings.** We selected LLaMA3-8B-Instruct (AI@Meta, 2024) and Qwen2.5-3B-Instruct (Team, 2024) as the policy models and Skywork-Reward-V2-Llama-3.1-8B (Liu et al., 2025) as the reward model for direct RL optimization.

**Hyperparameters.** For Qwen2.5-3B-Instruct, we set the learning rate to $6 \times 10^{-6}$ and the minimum weight coefficient for the original Reward Token Embedding to $\Omega = 0$. We set $K = 4$, and since $K$ also represents the number of times each query is reused, we used a total of only 3k queries. For

LLaMA3-8B-Instruct, we used a learning rate of $1 \times 10^{-6}$, set $\Omega = 0.5$ and the group size $K = 32$ and resulted in the use of only 0.375k queries.

### F.3 EXPERIMENTAL SETTINGS OF THE TL;DR TASK

We utilize the dataset trl-lib/TL;DR, sampling 2048 queries (i.e., $n = 2048$) from the training set at each step $t$, resulting in a total of 1000k trajectories for training. Due to the relatively short token length required for the summarization task, we limit the maximum number of generated tokens to 50 and perform RL optimization directly following the procedure in Algorithm 1.

After training, we used GPT-4 as the judge model (Zhang et al., 2024a; Rafailov et al., 2023; Zhu et al., 2025; Xie et al., 2025), taking the original summary content from the TL;DR dataset as the reference response, and calculated the win rate of the summaries generated by our trained policy model.

**LLM Settings.** Following prior work, we employ Pythia-2.8B-TL;DR-SFT , which has undergone supervised fine-tuning (SFT) on TL;DR, as the policy model, and Pythia-2.8B-TL;DR-RM , trained as a reward model on TL;DR, for direct RL optimization.

**Hyperparameters.** For policy model, we set the learning rate to $3 \times 10^{-6}$, the minimum weight coefficient for the original Reward Token Embedding $\Omega = 0$ and the group size $K = 4$.

### F.4 EXPERIMENTAL SETTINGS OF THE REWARD MODEL ACCURACY

In the dialogue task experiment, we retained the policy model $\pi_\theta$ and the reward model $r_\varphi$. We sampled $n_{total}$ preference pairs $\{x_i, y_{i,w}, y_{i,l}\}, i \in [n_{total}]$, from the test set of UltraFeedback, where $n_{total} = 1024$. When not using feedback from the policy, we computed $r_\varphi(x_i, y_{i,w})$ and $r_\varphi(x_i, y_{i,l})$, and counted the number of samples $n_{correct}$ where $r_\varphi(x_i, y_{i,w}) > r_\varphi(x_i, y_{i,l})$. The accuracy of the reward model was calculated as $acc_{r_\varphi} = n_{correct}/n_{total}$.

When incorporating policy feedback, we fed the chosen and rejected query-response pairs into the policy for a forward pass respectively and extracted the last layer's hidden states as policy feedback , denoted as $h_{i,w} = \pi_\theta(x_{i,w}, y_{i,w}) \in \mathbb{R}^{S_{i,w} \times D_p}$ and $h_{i,l} = \pi_\theta(x_{i,l}, y_{i,l}) \in \mathbb{R}^{S_{i,l} \times D_p}$, where $D_p$ denotes the policy model's hidden size, $S$ denotes the sequence length. Then, we calculated the accuracy based on the comparison between $r_\varphi(x_i, y_{i,w}, h_{i,w})$ and $r_\varphi(x_i, y_{i,l}, h_{i,l})$. We utilize the corresponding policy to provide feedback before and after the R2M pipeline.

### F.5 EXPERIMENTAL DETAILS OF THE PYTHIA-1B ON TL;DR TASK

We first present two examples of outputs from the policy model after reward hacking occurs, as shown below:

```
Completion 1:
'''
I(21M)needhelponhowto proceedwiththisgirlI'mseeing/sleepingwith(20F).
Iamafraidgettingtooinvolvedwillendinmegetting hurt.
<|endoftext|>[PAD][PAD][PAD][PAD][PAD][PAD][PAD][PAD][PAD][PAD][PAD][PAD]
'''

Completion 2:
'''
I[29M]amdatingmultiplepeople.HowdoInavigatethissituation?
Iamprettybadatdecipheringmyownemotions.
<|endoftext|>[PAD][PAD][PAD][PAD][PAD][PAD][PAD][PAD][PAD][PAD]
[PAD][PAD][PAD][PAD][PAD][PAD][PAD][PAD][PAD][PAD]
'''
```

We removed the spaces from a normal response and compared the reward values given by the reward model. The results are as follows:

```
Prompt:

'''User: SUBREDDIT: r/pettyrevenge
```

```
TITLE: So, my mom woke me up with a loud TV.

POST: She was in her living room, watching TV. This was at about 8:30
in the morning, and she was exercising. She turned the TV up extra loud
to hear it over her excercycle, and woke me up. I went in there asking
for her to turn it down. She said she didn't have to; I explained that I
always used headphones so she didn't have to deal with my noise and that
she should give me a little more respect, given that I paid rent at the
time.

She disagreed. I went back to my room, rather pissed off at the lack of
equality. I had no lock on my door; but I had a dresser right next to it,
so I pulled one of the drawers out enough so that it caused the door to
not be openable. Then, I turned my speakers up really loud and blasted
Gangnam Style on repeat, with the bass cranked up as high as it could go.

If you hate Gangnam Style for being overplayed, you will see why I chose
that particular song. I personally don't mind it. But here's the thing
about my bass; it vibrates the walls, making one hell of a lot of noise.
Needless to say, my mom was not pleased and shut off the internet. But it
was oh so worth it.

TL;DR:

Assistant:'''

Response without Space:

Momwokemeupwithloudmusic;
turnedoffinternet;pissedoffatlackofequality;
shutofftheinternetcompletelyafterblast
GangnamStyleonrepeatwithbassvibratingthewalls.

Token List:

['Mom', 'w', 'ok', 'eme', 'up', 'with', 'l', 'oud', 'music', ';',
'turned', 'off', 'intern', 'et', ';', 'p', 'iss', 'ed', 'off', 'atl',
'ack', 'o', 'fe', 'quality', ';', 'shut', 'off', 'the', 'intern',
'et', 'completely', 'after', 'blast', 'G', 'ang', 'nam', 'Style', 'on',
'repeat', 'with', 'b', 'ass', 'v', 'ibr', 'ating', 'the', 'walls', '.']

Reward: 3.366652488708496

Response with Spaces:

Mom woke me up with loud music; turned off internet; pissed off at lack
of equality; shut off the internet completely after blast Gangnam Style
on repeat with bass vibrating the walls.

Token List:

['Mom', 'Ġwoke', 'Ġme', 'Ġup', 'Ġwith', 'Ġloud', 'Ġmusic', ';',
'Ġturned', 'Ġoff', 'Ġinternet', ';', 'Ġpissed', 'Ġoff', 'Ġat',
'Ġlack', 'Ġof', 'Ġequality', ';', 'Ġshut', 'Ġoff', 'Ġthe', 'Ġinternet',
'Ġcompletely', 'Ġafter', 'Ġblast', 'ĠGang', 'nam', 'ĠStyle', 'Ġon',
'Ġrepeat', 'Ġwith', 'Ġbass', 'Ġvibr', 'ating', 'Ġthe', 'Ġwalls', '.']

Reward: 1.2537559270858765
```

It is evident that removing spaces from responses leads the reward model to assign higher scores. Due to the discrepancy between proxy and golden rewards, the reward model learns an implicit reward hacking pattern, preferring to assign higher scores to tokens not starting with "G". This has resulted

in a severe reward hacking phenomenon in our trained policy, where the policy tends to predict tokens without "G" to get high rewards, finally leading to the responses without spaces.

# G  MORE METHOD DETAILS OF R2M

## G.1  RLHF WORKFLOW

Here, We provide a detailed descrption of RLHF workflow.

**Supervised Fine Tuning.** RLHF typically begins with Supervised Fine Tuning (SFT), which involves training a pretrained language model in a supervised manner using high-quality, human-annotated dialogue examples. We denote the resulting model as $\pi_{\text{SFT}}$.

**Reward Modelling.** The second phase of RLHF involves learning a reward model to capture human preferences through annotated data $D = \{(x^i, y_w^i, y_l^i)\}_{i=1}^N$ where $y_w^i$ and $y_l^i$ denote the chosen and rejected responses to prompt $x^i$. The preferences are assumed to be generated by some unknown reward model $r^*(x, y)$ following the Bradley-Terry (BT) model (Bradley & Terry, 1952):

$$\mathbb{P}^*(y_w \succ y_l | x) = \frac{\exp(r^*(x, y_w))}{\exp(r^*(x, y_w)) + \exp(r^*(x, y_l))}.$$

Typically, a reward model $r_\varphi(x, y)$ is initialized from a pretrained LLM (usually $\pi_{\text{SFT}}$), with an additional projection layer (namely scoring head) $\phi : \mathbb{R}^{D_{rm}} \to \mathbb{R}^1$ added to map the last-layer hidden states of the final token $H_{\text{last}} \in \mathbb{R}^{D_{rm}}$ to a scalar reward $r_\varphi(x, y) = \phi(H_{\text{last}}) \in \mathbb{R}^1$. Since the rewards of query-response pairs are only related to $H_{\text{last}}$, we refer to it as the Reward Token Embedding.

Given the annotated preference data $D$, the reward model $r_\varphi$ is trained to assign higher reward to the chosen response $y_w$ compared to the rejected one $y_l$, by minimizing the negative log-likelihood under the BT model, where $\sigma$ denotes the sigmoid function:

$$\mathcal{L}(r_\varphi) = -\mathbb{E}_{(x,y_w,y_l)\sim D} \left[ \log \left( \sigma \left( r_\varphi(x, y_w) - r_\varphi(x, y_l) \right) \right) \right], \tag{7}$$

**RL Optimization.** The learned reward model $r_\varphi(x, y)$ is then employed to guide the RL policy optimization phase. Intuitively, the aim is to learn a policy $\pi_\theta$ that maximizes the reward $r_\varphi$ while not drifting too far away from $\pi_{\text{SFT}}$:

$$\max_{\pi_\theta} \mathbb{E}_{x\sim D, y\sim\pi_\theta} \left[ r_\varphi(x, y) \right] - \beta \mathbb{D}_{\text{KL}} \left[ \pi_\theta(y|x) \| \pi_{\text{SFT}}(y|x) \right], \tag{8}$$

where $\beta$ controls the deviation from the reference policy $\pi_{\text{SFT}}$, thus maintaining a balance between reward maximization and adherence to the SFT policy behavior.

## G.2  MOTIVATION OF LIGHTWEIGHT TRAINING

Although the computational overhead of the RL Optimization phase is primarily concentrated in the Trajectory Sampling phase, the computation cost of introducing a full reward model optimization phase remains unacceptable. Fortunately, the LLM component of the reward model has been trained on extensive text corpora, and with their large number of parameters, these models can develop generalizable representations, as demonstrated by Min et al. (2023); Wei et al. (2022); Brown et al. (2020); Lu et al. (2025). However, the learning of the projection weights $\phi$ in the reward model relies entirely on the preference data provided during reward model training. Consequently, the reliability of reward prediction is closely tied to the accuracy and generalizability of the projection weights. (Chen et al., 2020; Kirichenko et al., 2022; Riquelme et al., 2018; Xu et al., 2020)

Moreover, Kirichenko et al. (2022); Labonte & Muthukumar (2023); Lee et al. (2023) demonstrate that by freezing the network up to its last layer and retraining only the projection head with a smaller data set, it can greatly improve robustness of the neural network model.

These observations motivate us to freeze the LLM part of the reward model while updating only the parameters of the reward head.

