# OpenReview forum: "Reinforcement Reward Model with Policy Feedback"
_ICLR.cc/2026/Conference — ICLR 2026 Conference Withdrawn Submission_

### Official Review · Reviewer_Vyu8 · 2025-10-24

**Soundness:** 2
**Presentation:** 2
**Contribution:** 2
**Rating:** 2
**Confidence:** 3

**Summary:**

The submission proposes to address reward hacking by using not only the outputs of the policy in the reward model, but additionally using the hidden state of the policy as input for the reward model.
The authors propose a network structure that allows this and propose to fine-tune the reward model after each policy update with synthetic labels.
Experiments on ultrafeeedback and the tl;dr dataset show an improvement.

**Strengths:**

* preventing reward hacking is an important problem setting
 * The idea of using the hidden state (intermediate activations) instead of only the output of the policy is well motivated and could be helpful

**Weaknesses:**

* The main weakness is that the experiments do not sufficiently support the claim that the inclusion of the policy's hidden state indeed improves performance. The proposed approach includes two main parts: 1) add policy hidden state to RM input, 2) update RM with pseudo labels after each policy update. The ablations do not compare to the case of a normal RM, which is updated with pseudo labels after each policy update.
There is one ablation which replaces the policy information with random noise and performs worse, but inserting random noise into the RM hinders training and thus is not a fair comparison. Instead, comparing to a fixed input would make more sense, or instead simply comparing to an ablation that only finetunes the head of a standard RM.

* Experiments are not sufficiently rigorous. They are based on a single random seed each and, judging from Figure 5, training is not converged for either method. Hyper-parameters have seemingly not been optimized and are not fully reported.

* The writing is unclear at times, with undefined symbols, missing details and undefined metrics.

Minor issues:
 * Figure 1: The similarity measure is not specified. It is also not clear how the examples of reward hacking and not reward hacking were obtained.
 * "Hidden state" of the policy are never clearly defined. It seems to be the output of the last layer before the LM head, but this should be specified. This choice is also not
 * L92: pi_old is never used
 * L93: set X is not defined
 * L97: RLOO does not use groups, Hu 2025 is the correct citation for groups
 * L187: Reference to Section 2 is incorrect

**Questions:**

* Why is only the hidden state of the policy used, rather than the output token distribution or the intermediate activations? If we anthropomorphize the LM a bit, it seems reasonable that the "intention to reward hack" would arise in earlier or intermediate layers, not in the final layer.
 * How was Figure 1 created?

---

### Official Review · Reviewer_4wJU · 2025-10-27

**Soundness:** 3
**Presentation:** 3
**Contribution:** 2
**Rating:** 4
**Confidence:** 4

**Summary:**

The paper targets reward hacking in RLHF by proposing R2M, a lightweight framework that redesigns the reward model’s scoring head to incorporate the hidden states of the current policy model. The reward model is trained with pairwise preferences using a Bradley–Terry objective. Experimental results show the effectiveness of the proposed method.

**Strengths:**

1. Addresses an important practical issue in RLHF (reward hacking) with a relatively simple architectural change.

2. Integrating policy hidden states into the reward scorer is an interesting and potentially useful idea that could improve reward-model calibration to the policy’s distribution.

**Weaknesses:**

1. Dependence on Bradley–Terry preference modeling: While pairwise preference RMs are standard, the paper does not situate R2M against verifiable reward approaches that are increasingly dominant for math and code. These verifiable rewards are often lighter-weight by grounding the reward in external correctness.

2. Computational overhead: The method requires modifying the reward model and iteratively training both the policy and reward models. It is unclear whether the coupling increases training cost, instability, or maintenance complexity compared to standard RLHF pipelines. A clear accounting of compute, memory, and wall-clock time, plus sample efficiency, is missing.

3. Limited scenarios and benchmarks: Evaluation is restricted to dialogue and TL;DR, without coverage of standard open benchmarks for LLM evaluation (e.g., MT-Bench, Arena-Hard, AlpacaEval 2, Opencompass).

4. Metrics do not directly measure reward hacking: Reported metrics seem to focus on reward-model scores or internal alignment proxies, making it hard to substantiate claims about reduced reward hacking.

**Questions:**

Multi-reward scenario:

1. How does R2M extend to multiple objectives (e.g., helpfulness, harmlessness, calibration, style) or task-specific rewards (math, code correctness, factuality)?

2. Can the scoring head be designed to aggregate multiple reward signals (scalarization, mixture-of-experts, or listwise ranking) while preserving robustness against hacking?

Benchmark coverage:

1. How R2M performs on public benchmarks such as arena

2. How does R2M perform relative to verifiable reward on math and code tasks?

3. How robust are the results across different base models, training scales, and domains?


Measuring reward hacking:

1. How is “reward hacking” defined in this work?

2. What metrics directly quantify reward hacking beyond reward-model scores?

---

### Official Review · Reviewer_msVd · 2025-10-31

**Soundness:** 2
**Presentation:** 2
**Contribution:** 3
**Rating:** 4
**Confidence:** 5

**Summary:**

The paper analyzes policy model forward hidden states for responses with and without reward hacking and finds notable differences. Building on this finding, it proposes R2M, a reward model that takes both the response and the policy model’s hidden states as inputs to mitigate reward hacking. The method further introduces a Group Reward Entropy Bradley–Terry loss to iteratively update R2M. Experiments on *AlpacaEval2* and *TL;DR* report performance gains.

**Strengths:**

1. The motivation analysis is sound. Studying reward hacking through the lens of policy hidden states is novel.
2. The R2M architecture leverages policy hidden states to alleviate reward hacking, and its effectiveness is validated on *AlpacaEval2* and *TL;DR*.

**Weaknesses:**

#### Motivation
1. The identified reward-hacking patterns need more quantitative evidence. Do the conclusions hold at larger scales?

2. How do reward hacking type and prompt type relate to *similarity matrix*? If the same prompt triggers both length hacking and formatting hacking, do hidden states remain similar?

3. Does the *similarity matrix* of hidden states change under policy distribution shift? As training progresses, do the conclusions still hold?

#### Method
1. The method uses a *bootstrap* scheme to train/update the reward model. On line 242, “we select only the samples with the highest and lowest scores...” Considering that when reward hacking occurs, the reward of the response that shows reward hacking will be higher than that of the normal response, this selection may cause the optimization in **Equation (4)** to push in the wrong direction.

2. The method *pre-sets* the number of training rounds $ T $. In practical RL, convergence rounds are uncertain; it is difficult to determine $T$ *a priori*, making it hard to judge whether the key R2M component is actually taking effect. More *ablation* on $T$ is recommended.

3. The crucial *Sequence-to-Token Cross-Attention* in R2M lacks a formal, equation-level definition.

#### Baselines
1. The evaluation benchmarks are limited; broader assessments on MT-Bench, Arena-Hard, IFEval, etc., would be more convincing.
2. In Table 2, R2M is compared *against* RLOO, GRPO, ReMax, etc. Given the claim that R2M can be plugged into **any REINFORCE-based RLHF framework**, a more appropriate comparison is to apply R2M within RLOO, GRPO, ReMax, etc., and compare R2M-augmented vs normal reward model within each framework.

### Experiments
1. The experiments on reward hacking are insufficient and not fully quantitative. Since the paper’s main claim is to mitigate reward hacking, experiments should focus more on this—beyond space hacking—including length hacking, pattern/template hacking, and others.
2. In Table 4, the WR (%) of RLOO is significantly lower than SFT, which undermines a fair comparison and may indicate RL was not conducted properly.
3. The computational cost analysis needs to be more formal. In practice, jointly updating the policy and the reward model can make convergence difficult. The paper should report the number of steps to converge for a fair comparison.

### Typos
1. Line 246 (Eq. 4): $y_{i,w}$ and $h_{i,w}$ — the index $w$ is undefined.
2. Line 252: *foward* → *forward*.
3. Line 334: *increse* → *increase*.

**Questions:**

1. R2M passes the policy model’s hidden state to the reward model, but there is no explicit signal indicating whether that hidden state corresponds to a hacked response. How does R2M learn to distinguish  hacked cases?

---

### Official Review · Reviewer_2sd2 · 2025-11-08

**Soundness:** 2
**Presentation:** 2
**Contribution:** 2
**Rating:** 4
**Confidence:** 4

**Summary:**

The paper introduces policy feedback into reward scoring stage. Here, policy feedback refers to taking the hidden state embedding of the policy and cross attends to such embeddings from the reward computation. When reward is adapted this way they show performance improvement during the policy optimization process.

**Strengths:**

The paper is interesting in that it introduces much richer interactions between policy and reward, allowing for policy to pass on richer info into reward computation, and they show improvements over a few baselines. The idea itself is somewhat novel.

**Weaknesses:**

I think the paper lacks in a few important aspects such as technical solidity, presentation clarity etc which I will detail in the questions below.

**Questions:**

==== **algorithm 1** ====

I am a bit confused by the logic in algorithm 1, where it seems that we initially sample and generate scores $y_i$ from the RM and later will regress RM against the very same reward labels $y_i$ but using updated hidden state from the policy. I suppose this is the adaptive strategy highlighted in this work? i.e., not only is there an architectural improvement (cross attention to hidden state), there is also an online adaptation to ensure RM is in sync with policy.

I worry that this updating process will be a bit sensitive to hypers related to training the RM online because it seems that too strong of a learning rate might erase the original RM's information. After all it's a form of self-distillation with labels from a previous version of the RM itself, so I think it is useful to ablate and show how sensitive this process is to various hyperparameters used for updating RM.

==== **an architectural alone baseline** ====

Related to the above I think another important baseline is, what if you don't do the online adaptation and simply just do the cross attention using the policy feedback. This is already a form of online adaptation but does not require updating and self-distilling RM online.

I think another baseline is to train and update the RM once at the beginning of policy training, and keep its weights fixed throughout. This experiment will let us know whether fully online adaptation is necessary.

Overall I feel I am not sure where the performance gains are because of so many changes all at once, and it is useful to ablate each part to elicit where the true performance gains lie and if some parts of the algorithm are not necessary (in light of improvement in Fig 4).

==== **Fig 4 and mechanisms behind the improvement** ====

I am generally intrigued but meanwhile a bit surprised at the improvements in Fig 4. I think overall it's not clear why the improvements are possible in the first place simply because of the adaptation proposed in the paper. Indeed, there is more feedback from policy to RM and hidden state cross attention allows for much richer information, but at the end the RM distills back into its own labels and it's not clear to me new information has been distilled into the policy. Maybe the cross attention somehow gives more info to the RM? But this does not provide new source of "ground truth" to the RM itself, so where do the gains come from intuitively? Does it come from a smoother optimization landscape from the RM? Does it come from a different effective hyper-parameter from the RL run?

I don't know answers to the above but I think more ablations are warranted for a better understanding. Just putting together a stack of system with performance the source of gains is good but understanding is more important.

---

### Note · Authors · 2026-01-08

I have read and agree with the venue's withdrawal policy on behalf of myself and my co-authors.